



# Clustering of settling microswimmers in turbulence

Jingran Qiu[1], Zhiwen Cui[1], Eric Climent[2], and Lihao Zhao[1]

[1]AML, Department of Engineering Mechanics, Tsinghua University, 100084 Beijing, China
[2]Institut de Mécanique des Fluides de Toulouse (IMFT), UMR5502 Université de Toulouse, CNRS. Allée du Prof. Camille Soula – 31400 Toulouse, France

**Correspondence:** Lihao Zhao (zhaolihao@mail.tsinghua.edu.cn)

**Abstract.** Clustering of plankton plays a vital role in several biological activities including feeding, predation and mating. Gyrotaxis is one of the mechanisms that induces clustering. A recent study (Candelier et al., 2022) reported a fluid inertial torque acting on a spherical micro-swimmer, which is analogous to a gyrotactic torque. In this study, we model plankton cells as micro-swimmers that are subject to gravitational sedimentation as well as a fluid inertial torque. We use direct numerical simulations to obtain the trajectories of swimmers in homogeneous isotropic turbulence, and investigate their clustering by Voronoï analysis. Our findings indicate that fluid inertial torque leads to notable clustering, with its intensity depending on the swimming and settling speeds of swimmers. By Voronoï analysis, we demonstrate that swimmers preferentially sample downwelling regions where clustering is more prevalent.

## 1 Introduction

Plankton are known to form small scale clusters in turbulent environment (Rothschild and Osborn, 1988). These clusters can be down to centimeter-scale and significantly impact basic life processes of plankton such as feeding, predation and mating. Gyrotaxis is one of the mechanisms that causes plankton to form clusters. Many plankton experience a gravitational stabilizing torque that cause them to swim against gravity (Kessler, 1986). When plankton encounter flow shear, the gyrotactic torque opposes the fluid viscous torque and tends to stabilize the swimming direction of the plankton (Qiu et al., 2022b).

Gyrotactic plankton can form different kinds of clustering depending on the flow characteristics. For instance, plankton accumulates in the center or the wall regions in downward or upward pipe flow, respectively (Kessler, 1985). Plankton that are vertical migrating also form clustering when they encounter a shear layer that interrupts the migration (Durham et al., 2009). Plankton in turbulence form small scale clusters that can be characterized by the swimming speed and the intensity of gyrotactic torque. Durham et al. (2013) modeled plankton as spherical gyrotactic micro-swimmers and numerically studied their fractal clustering in homogeneous isotropic turbulence. They demonstrated that the intensity of clustering depends on the swimming speed and the intensity of gyrotaxis that is typically characterized by the inverse of a timescale $B$. Clustering is also shown to be correlated to the preferential sampling of downwelling regions (Durham et al., 2013). Later, Zhan et al.





(2014) numerically investigated the effect of plankton shape on the clustering. Elongated swimmers are more sensitive to fluid

shear than spherical ones, weakening the clustering of strongly gyrotactic swimmers. However, elongation causes preferential alignment in local fluid structures, strengthening the clustering of weakly gyrotactic swimmers. To further clarify the complex relationship between clustering and the swimming speed, gyrotaxis and shape of the swimmers, Gustavsson et al. (2016); Fouxon and Leshansky (2015) established the theory of cluster using stochastic models. These theories were later verified by direct numerical simulations of swimmers in homogeneous isotropic turbulence (Borgnino et al., 2018).

Previous studies suggested that gyrotaxis originates from the asymmetric body structures, such as nonuniform mass distribution (bottom-heaviness) (Kessler, 1985, 1986; Pedley and Kessler, 1987). However, a recent study by Candelier et al. (2022) modeled planktonic microorganisms as settling spherical squirmers and found that a fluid inertial torque drives the squirmer to swim against gravity. The squirmer model is proposed by Lighthill (1952) and improved by Blake (1971) to describe the slip velocity on the surface of microorganisms generated by the movement of cilia. The squirmer model can describe the typical

propulsion modes such as puller for algae and pusher for *E. coli* by changing model parameters. Both theory and simulations indicated that fluid inertial torque on a settling squirmer is analogous to a gyrotactic torque, with a magnitude that is proportional to the settling and swimming speeds (Candelier et al., 2022). Planktonic organisms are usually slightly negatively buoyant, thus subject to a gravitational settling effect. For instance, dinoflagellates have a typical swimming speed of 300 μm/s and settling speed of 30 μm/s (Smayda, 2010). Larger organisms such as copepod nauplii have swimming speeds up to 1000 μm/s and

settling speeds of 200 μm/s (Titelman and Kiørboe, 2003). As pointed out by Candelier et al. (2022), an organism with large swimming and settling speeds obtain a fluid inertial torque that is comparable to typical gyrotactic torque. However, earlier studies usually neglected the gravity sedimentation and the fluid inertial torque, highlighting the need to consider their effects on the motion of swimming, settling plankton.

   In this study, we aim to analyze the clustering of planktonic swimmers under the influence of fluid inertial torque. We model

plankton as point-like spherical micro-swimmers undergoing gravity sedimentation. We use direct numerical simulations of swimmer trajectories in homogeneous isotropic turbulence to analyze their clustering characteristic. In section 2.1, we describe the model and the numerical approaches. In section 3, we investigate the clustering using Voronoï analysis and show the relation between clustering and preferential sampling of downwelling regions. In section 4, we draw the conclusions of the present study.

## 2 Methods

### 2.1 Model of spherical swimmers

In the present study, we consider a spherical swimmer undergoing gravitational sedimentation as shown in Figure 1. The motion of plankton in fluid flows is usually described by a micro-swimmer model (Durham et al., 2009, 2013; Gustavsson et al., 2016; Lovecchio et al., 2019; Zhan et al., 2014), which assumes a plankton to be a point-like micro-swimmer carried by a fluid flow



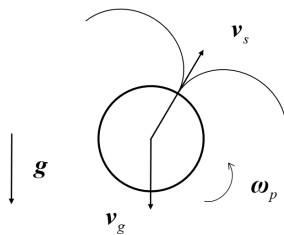

**Figure 1.** A sketch of a settling swimmer.

whose scales are much larger than the plankton body length. The dynamics of the swimmer is governed by

$$m_p \frac{\mathrm{d}\boldsymbol{v}}{\mathrm{d}t} = 6\pi a \gamma \rho_f (\boldsymbol{u} - \boldsymbol{v}) + m_p (1 - \frac{\rho_f}{\rho_p})\boldsymbol{g} + F_s \boldsymbol{n} \tag{1}$$

$$m_p I_p \frac{\mathrm{d}\boldsymbol{\omega}_p}{\mathrm{d}t} = 6\pi a \rho_f \gamma C \left( \frac{1}{2}\boldsymbol{\omega} - \boldsymbol{\omega}_p \right)$$
$$+ \frac{9 m_p \rho_f}{8\rho_p} \left[ (\boldsymbol{v} - \boldsymbol{u}) \times \boldsymbol{v}_s \right], \tag{2}$$

where $m_p$, $\rho_p$ and $a$ are the mass, the density and the radius of the swimmer, respectively. Eq. (1) governs the translational motion of the swimmer, where the first term on the right-hand-side denotes the Stokes drag. Here, $\rho_f$, $\gamma$ and $\boldsymbol{u}$ are the density, kinematic viscosity, and velocity of fluid, respectively, and $\boldsymbol{v}$ denotes the velocity of the swimmer. The second term represents the gravity force and buoyancy on the swimmers due to gravity acceleration $\boldsymbol{g}$. The third term represents a swimming force $F_s$ in the direction of the head of swimmer, denoted as $\boldsymbol{n}$. Meanwhile, Eq. (2) governs the rotation of the swimmer, where $I_p = 2a^2/5$ denotes the moment of inertia per unit mass, and $\boldsymbol{\omega}_p$ represents the angular velocity of the swimmer. The first term on the right-hand-side of Eq. (2) represents the Jeffery torque (Jeffery, 1922), where $C = 4a^2/3$, and $\boldsymbol{\omega}$ is the vorticity of the fluid flow. The second term represents the fluid inertial torque experienced by a squirmer (Candelier et al., 2022), where $\boldsymbol{v}_s$ represents the swimming speed of the squirmer in a quiescent fluid.

Using a velocity and a timescale of the flow $u_f$ and $\tau_f$, we make Eqs. (1) and (2) dimensionless,

$$St \frac{\mathrm{d}\boldsymbol{v}'}{\mathrm{d}t'} = \boldsymbol{u}' - \boldsymbol{v}' + \Phi_s \boldsymbol{n} + \Phi_g \boldsymbol{e}_g, \tag{3}$$

$$St \frac{I_p}{C} \frac{\mathrm{d}\boldsymbol{\omega}_p'}{\mathrm{d}t'} = \frac{1}{2}\boldsymbol{\omega}' - \boldsymbol{\omega}_p' + \frac{3\tau_f u_f^2}{16\gamma} \left[ (\boldsymbol{u}' - \boldsymbol{v}') \times \boldsymbol{v}_s' \right], \tag{4}$$

where the quantities with primes are dimensionless. In above equations, the Stokes number $St = (2a^2 \rho_p)/(9\gamma \rho_f \tau_f)$ reflects the inertial of the swimmer relative to the fluid of the same mass. $\Phi_s = v_s/u_f$ and $\Phi_g = v_g/u_f$ are the dimensionless swimming and settling speeds, respectively. According to typical plankton parameters (Qiu et al., 2022a), the inertia of plankton is



negligible and $St \to 0$. In such limit, the dynamics (3) and (4) can be simplified

$$\frac{\mathrm{d}\boldsymbol{x}'}{\mathrm{d}t'} = \boldsymbol{v}', \tag{5}$$

$$\frac{\mathrm{d}\boldsymbol{n}}{\mathrm{d}t'} = \boldsymbol{\omega}'_p \times \boldsymbol{n}, \tag{6}$$

$$\boldsymbol{v}' = \boldsymbol{u}' + \Phi_s \boldsymbol{n} + \Phi_g \boldsymbol{e}_g, \tag{7}$$

$$\boldsymbol{\omega}'_p = \frac{1}{2}\boldsymbol{\omega}' + \frac{1}{2\Psi_I}\left(\boldsymbol{e}_g \times \boldsymbol{n}\right). \tag{8}$$

where $\Psi_I = 8\gamma/(3\tau_f u_f^2 \Phi_s \Phi_g)$. Note that the second term on the right hand side of Eq. (8) is analogous to the gyrotactic effect induced by bottom-heaviness, which is typically expressed as $(2\Psi)^{-1}(\boldsymbol{e}_g \times \boldsymbol{n})$ (Kessler, 1986; Durham et al., 2013). This torque is quantified by a dimensionless reorientation timescale $\Psi = B/\tau_f$, where $B$ denotes the time required for a swimmer under gyrotactic torque to restore upward orientation from an inclined orientation in still fluid. Eq. (8) indicates that fluid inertial torque on a squirmer swimmer provides effective gyrotaxis with a dimensionless reorientation timescale $\Psi_I$.

In turbulence, we can take the turbulence Kolmogorov velocity and timescales $u_\eta$ and $\tau_\eta$ as the characteristic scales of the flow. Using the relation $\gamma = u_\eta^2 \tau_\eta$, $\Psi_I$ can be simplified as

$$\Psi_I = \frac{8}{3\Phi_s \Phi_g}. \tag{9}$$

The typical value of $\Phi_s$ and $\Phi_g$ of plankton can be estimated with their swimming and settling speeds as well as the Kolmogorov velocity scale of ocean turbulence. As summarized in Qiu et al. (2022a), the swimming speeds of different species vary from 200 to 1500 μm/s, and the settling speeds vary from 10 to 200 μm/s. The Kolmogorov velocity scale of ocean turbulence can be estimated from the typical dissipation rate $\epsilon = 10^{-9}$ to $10^{-6}\mathrm{m^2 s^{-3}}$ (Kiørboe and Enric, 1995), yielding $u_\eta = (\gamma\epsilon)^{1/4} = 178$ to $1000$ μm/s with $\gamma = 10^{-6}$ m$^2$s$^{-1}$. Based on these estimations, we consider the typical parameter space of $0 < \Phi_s < 10$ and $0 < \Phi_g < 1$.

### 2.1.1 Direct numerical simulations of swimmers in turbulence

The motion of swimmers in homogeneous isotropic turbulence is simulated by a Eulerian-Lagrangian direct simulations. The flow field is resolved in the Eulerian frame, while the motions of individual swimmers are solved along the Lagrangian trajectories using local flow information at swimmers' positions. The incompressible turbulent flow is directly simulated by solving the Navier-Stokes equations:

$$\frac{\partial \boldsymbol{u}}{\partial t} + \boldsymbol{u} \cdot \nabla \boldsymbol{u} = -\frac{\nabla p_{\mathrm{f}}}{\rho_f} + \gamma\nabla^2\boldsymbol{u} + \boldsymbol{f}, \tag{10}$$

$$\nabla \cdot \boldsymbol{u} = 0, \tag{11}$$

where $p_{\mathrm{f}}$ is the pressure of fluid. An external force $\boldsymbol{f}$ is applied to sustain turbulence and balance the rate of viscous dissipation at the Kolmogorov scale $\eta$. The force is applied to the large scale motion using the scheme proposed by Machiels (1997).





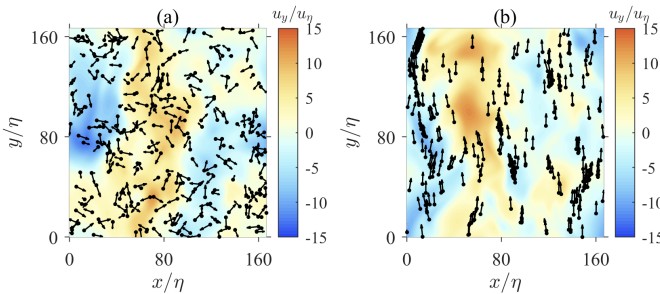

**Figure 2.** Instantaneous spatial distribution of swimmers in homogeneous isotropic turbulence. Black dots and tiny arrows represent the position and swimming direction of each swimmer, respectively. Background contour represents the vertical fluid velocity $u_y$. (a) Non-settling swimmers ($\Phi_g = 0$, $\Phi_s = 10$). (b) Settling swimmers ($\Phi_g = 1$, $\Phi_s = 10$).

Periodic boundary conditions are applied on all boundaries of the cubic domain with a size of $(2\pi)^3$. We use pseudo-spectral method to solve the Navier-Stokes equations, and we adopt the 3/2 rule for reducing the aliasing error on the nonlinear term. The separation between turbulent motion of large and small scales is quantified by the Taylor-Reynolds number $\mathrm{Re}_\lambda = u_{\mathrm{rms}} L_\lambda / \gamma$, where $u_{\mathrm{rms}}$ is the root-mean-square velocity, and $L_\lambda = u_{\mathrm{rms}} \sqrt{15\gamma\epsilon^{-1}}$. In the present study, we consider a turbulence of

$\mathrm{Re}_\lambda = 60$. To resolve the turbulent flow down to the Kolmogorov scale, we use $96^3$ grid points, which allows a maximum wave number resolved to be 1.78 times greater than the Kolmogorov wave number to ensure the accuracy of resolution even at Kolmogorov scales (Pope, 2000). The initial flow field is set as a random flow with an exponential energy spectrum, and an explicit second-order Adams-Bashforth scheme is used for time integration of Eqs. (10) and (11) with a time step smaller than $0.01\tau_\eta$ (Rogallo, 1981).

Swimmers are initialized with random positions and orientations after turbulence is fully developed. When solving the trajectories of swimmer, fluid velocity and its gradients at Eulerian grid points are interpolated by a second-order Lagrangian method at the positions of swimmers. Eqs. (5) and (6) are integrated by the same second-order Adams-Bashforth scheme as the fluid phase. For each parameter configuration, $10^5$ swimmers are simulated and the statistics are obtained by making an ensemble average over more than 80 uncorrelated time samples after the dynamics has reached a steady state.

**3   Results**

The instantaneous location and orientation of swimmers are depicted in Figure 2. When swimmers are not settling (Figure 2a), they are distributed randomly with random orientation. Spherical swimmers are known to exhibit random orientation due to the random fluid vorticity of turbulence. As a result, their motions in turbulence remain random and no cluster is formed. However, when swimmers are settling under the influence of the gravity (Figure 2b), they tend to swim upwards and form clusters due

to the contribution of fluid inertial torque as predicted by Candelier et al. (2022). As discussed earlier, the fluid inertial torque on a settling swimmer induces an effective gyrotaxis mechanism. Gyrotactic swimmers are known to form spatial clusters and preferentially sample regions with downwelling or upwelling fluid velocity. Previous studies have documented that these

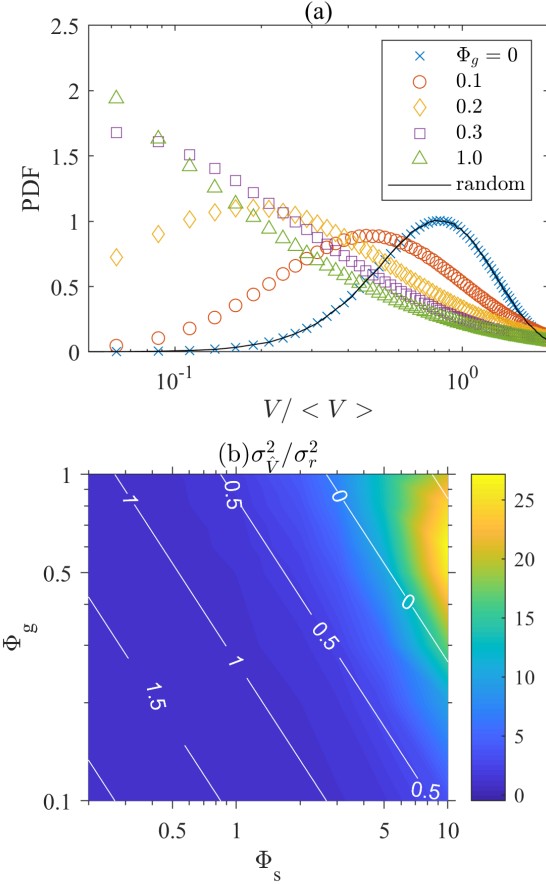

**Figure 3.** (a) Probability distribution function (PDF) of the volumes of Voronoï cells, normalized by the mean volume $\langle V \rangle$. $\Phi_s = 10$. (b) Variance of Voronoï volumes $\sigma_{\hat{V}}^2 = E(V/\langle V \rangle - 1)^2$ normalized by the value of randomly distributed particles. The white contour lines represent the value of $\log_{10} \Psi_I$ in the parameter space.

phenomena depend on the swimming speed, reorientation time, and the shape of swimmers (Durham et al., 2013; Zhan et al., 2014; Gustavsson et al., 2016; Borgnino et al., 2018). Here, gyrotaxis is induced by fluid inertial torque with a reorientation time

quantified by $\Psi_I$, which depends on the swimming and settling speeds of swimmers. $\Psi_I$ cannot be treated as an independent parameters as earlier studies did (Durham et al., 2013; Zhan et al., 2014; Gustavsson et al., 2016; Borgnino et al., 2018). Hence, the picture of clustering may differ from previous studies, and it is worth a further investigation.

## 3.1 Clustering

The clustering of swimmers is quantified by a three-dimensional Voronoï tessellation (Nilsen et al., 2013; Monchaux et al.,

2010). The whole domain is divided into many Voronoï polyhedrons based on the positions of swimmers, with each polyhedron





containing one swimmer. Any point in a polyhedron is closest to the corresponding swimmer among all swimmers. The volume of a Voronoï polyhedron is smaller when the corresponding swimmer is surrounded by more other swimmers, and vice versa. Therefore, the distribution of Voronoï polyhedron volumes quantifies the clusters of swimmers.

We use the MATLAB toolbox 'voronoi.m' and 'convhull.m' to compute the vertices of Voronoï polyhedrons and calculate their volumes. Figure 3(a) shows the probability distribution function (PDF) of Voronoï volumes for swimmers with different settling speeds. The PDF of Voronoï volumes of non-settling swimmers remains the same as the one generated from random positions, indicating the absence of clustering. When settling speed increases, the PDFs becomes skewed and a peak at small $V/\langle V \rangle$ appears. This indicates the occurrence of clustering, because swimmers in clusters remain close to each other and their Voronoï volumes are thus small. Settling swimmers form clusters due to the effect of fluid inertial torque. As shown in Eq. (8), the fluid inertial torque drives settling swimmer to orientate upward with a finite reorientation timescale $\Psi_I$. This is analogous to the effect of bottom-heaviness (Kessler, 1986), which also drives swimmers to orientate upward with a timescale $\Psi$ dependent on the offset of the center of gravity with the center of hydrodynamic forces. For inertial torque, however, the timescale $\Psi_I$ is inversely proportional to both the settling and swimming speeds of the swimmer.

To show how clustering depends on the settling and swimming speeds, in Figure 3(b) we depict the variance of Voronoï volumes for different $\Phi_g$ and $\Phi_s$. The corresponding magnitude of $\log_{10} \Psi_I$ is also shown by white contour lines. The variance of Voronoï volumes quantifies the intensity of clustering because a stronger clustering results in a more nonuniform distribution of Voronoï volumes with larger variance. The results show that clustering becomes stronger with increasing $\Phi_s$ and $\Phi_g$, and reaches a peak at $\Phi_g \approx 0.5$ and $\Phi_s \approx 10$. Further increasing $\Phi_g$ leads to a drop of the clustering intensity. This trend can be explain using the dimensionless reorientation timescale $\Psi_I$, which is inversely proportional to $\Phi_s$ and $\Phi_g$ (Eq. 9). When $\Psi_I$ is zero, gyrotaxis is infinitely strong, causing swimmers to swim straight up against gravity, yielding $\boldsymbol{n} = -\boldsymbol{e}_g$. Since the fluid is incompressible, according to Eq. (7), the velocity field of swimmers has zero divergence, $\nabla \cdot \boldsymbol{v} = \nabla \cdot \boldsymbol{u} = 0$, indicating that no clustering is formed. When $\Psi_I$ is infinitely large, the fluid inertial torque is negligible, and the swimming direction is entirely determined by turbulent shear and becomes random, resulting in no clustering. Therefore, the maximal clustering is expected to occur at a finite $\Psi_I$. Durham et al. (2013) observed that intensity of clustering of gyrotactic swimmers reaches its maximal when $\Psi$ is of the order of unity (Durham et al., 2013). Since $\Psi_I$ is analogous to $\Psi$, the maximal clustering in the present case is also observed at certain $\Phi_s$ and $\Phi_g$ that yields $\Psi_I \sim 1$.

## 3.2 Preferential sampling of downwelling regions

The clustering of spherical gyrotactic swimmers in turbulence has been shown to be associated with preferential sampling of downwelling regions (Durham et al., 2013). Figure 4 shows the mean vertical fluid velocity at the position of swimmers. Swimmers always sample downwelling regions, and the maximal sampling occurs at large $\Phi_s$ but moderate $\Phi_g$ which yields $\Psi_I \approx 1.0$. This observation is similar to Durham et al. (2013) where the maximal preferential sampling is also reached when $\Psi \approx 1$.

Comparing Figure 4 and Figure 3(b), we observed a very similar trend between the sampling of downwelling regions and the intensity of clustering. The magnitude of both quantities increase with $\Phi_s$ and reach their maximal at a large $\Phi_s$ and a moderate



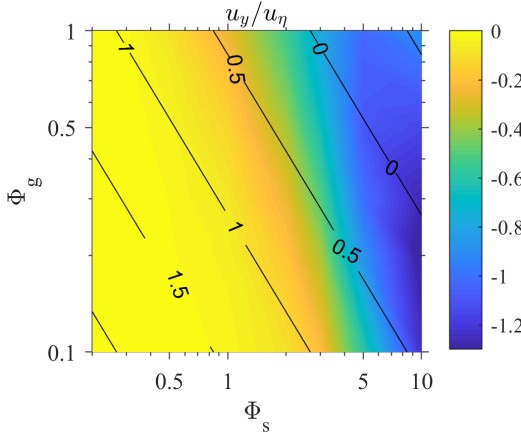

**Figure 4.** (a) Mean vertical fluid velocity at swimmers' positions, $\langle u_y \rangle$, normalized by $u_\eta$ as a function of $\Phi_g$ and $\Phi_s$. The black contour lines represent the value of $\log_{10} \Psi_I$ in the parameter space.

$\Phi_g$. This supports the theory that clustering occurs in downwelling regions (Durham et al., 2013; Fouxon and Leshansky, 2015; Gustavsson et al., 2016). Durham et al. (2013) showed that the divergence of the swimmer velocity field $\nabla \cdot \boldsymbol{v} \propto -\nabla^2 u_y$. Since the $\nabla^2 u_y$ is negatively correlated to $u_y$ in incompressible, homogeneous isotropic turbulence, the sinks of swimmer velocity field tend to be located in downwelling regions with $u_y < 0$. Here, we provide more direct evidence for the clustering in downwelling regions.

Voronoï analysis allows us to track the Voronoï volume of each swimmer. Based on the values of volumes, we can distinguish whether each swimmer is inside a cluster (with small Voronoï volume) or moving alone away from other swimmers (with large Voronoï volume). Figure 5 shows the joint probability distribution function (joint PDF) of $u_y$ and $\log(V/\langle V \rangle)$ for swimmers with different settling speeds. When $\Phi_g = 0$ (Figure 5(a)), fluid inertial torque vanishes and swimmers do not preferentially sample downwelling regions, resulting in a symmetric joint PDF with respect to $u_y = 0$. Moreover, because non-settling swim-

mers do not form clusters and their Voronoï volumes tend to be uniformed, the joint PDF along $\log(V/\langle V \rangle)$ is concentrated at the peak. However, when $\Phi_g > 0$, the joint PDF becomes asymmetric with respect to $u_y$ (Figure 5(b)). The peak shifts towards $u_y < 0$ because swimmers preferentially sample downwelling regions. Moreover, $\log(V/\langle V \rangle)$ tends to be smaller when $u_y < 0$, indicating that swimmers in downwelling regions are more likely to form clusters. When settling speed increases to $\Phi_g = 0.5$ (Figure 5(c)), the joint PDF becomes flattened along $\log(V/\langle V \rangle)$, because the intensity of clustering reaches its

maximal (see Figure 3(b)), making it more probable for swimmers to have both smaller and larger Voronoï volumes. Furthermore, the joint PDF becomes less asymmetric with respect to $u_y$, indicating that strong clustering no longer occurs only in downwelling regions. When $\Phi_g$ further increases to $\Phi_g = 1$, the distribution becomes slightly concentrated again because the intensity of clustering is weakened compared to the case of $\Phi_g = 0.5$. In general, the joint PDFs reveal that swimmers are more likely to form cluster in downwelling regions, but when clustering is intense, the bias is weak.





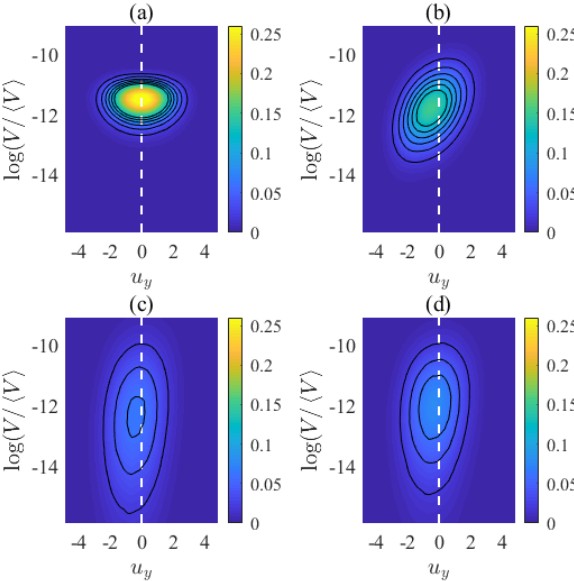

**Figure 5.** Joint probability distribution function (PDF) of vertical fluid velocity $u_y$ and the Voronoï volumes $\log(V/\langle V\rangle)$. $\Phi_s = 10$ for all panels. White dashed lines correspond to $u_y = 0$. (a) $\Phi_g = 0$. (b) $\Phi_g = 0.2$. (c) $\Phi_g = 0.5$. (d) $\Phi_g = 1.0$.

## 4  Conclusions

A settling spherical squirmer experiences a fluid inertial torque that causes it to swim against gravity, acting as an effective gyrotactic torque (Candelier et al., 2022). While previous studies have focused on gyrotactic torque originating from bottom-heaviness, the role of fluid inertial torque has been neglected (Durham et al., 2013; Zhan et al., 2014; Gustavsson et al., 2016; Borgnino et al., 2018). In the present study, we modeled the inertia-less micro-swimmer under the influence of fluid inertial torque. The magnitude of the torque is quantified using a dimensionless reorientation timescale $\Psi_I$ which is proportional to the inverse of dimensionless swimming speed ($\Phi_s$) and settling speed ($\Phi_g$).

Using direct numerical simulation, we investigated the clustering of swimmers under fluid inertial torque. We quantified the clustering using a Voronoï analysis. When swimmers are not settling, the fluid inertial torque vanishes, and the swimmers are randomly distributed resulting from a random direction of swimming, with no clustering observed. Settling swimmers experience a fluid inertial torque and behave similarly to gyrotactic swimmers. We observed that swimmers form more intense clustering when $\Phi_s$ and $\Phi_g$ become larger, with maximal clustering intensity occurring at the largest $\Phi_s$ and a modest $\Phi_g$, corresponding to $\Psi_I \sim 1$.

We also examined how the clustering of spherical swimmers is related to their preferential sampling of downwelling regions. We found that when swimmers are not settling, their dynamics remains isotropic, and no preferential sampling is observed in the gravity direction. However, the fluid inertial torque, as well as the settling speed, break this symmetry, and drive settling swimmers to sample downwelling regions. The sampling is more pronounced with larger $\Phi_s$ and $\Phi_g$, reaching the maximum

when $\Psi_I \approx 1$. The trend of preferential sampling shows a similar pattern to that of clustering intensity, indicating a correlation between the two phenomena. We used the joint PDF of Voronoï volumes and local vertical fluid velocity to demonstrate that swimmers tend to form clusters in downwelling regions.

The fluid inertial torque on settling swimmers acts like a gyrotactic torque and can cause the formation of small-scale clusters, highlighting the importance of fluid inertial effects on the dynamics of plankton. However, most earlier studies did not consider gravitational sedimentation, leading to the neglect of fluid inertial torque. This underestimates the intensity of gyrotaxis because the total gyrotactic torque is contributed by both fluid inertial torque and bottom-heaviness. In addition, the fluid inertial torque is proportional to the swimming and settling speeds, making the gyrotaxis reorientation time a dependent

parameter. Therefore, planktonic swimmers have the potential to tune their gyrotaxis and clustering intensity by adjusting their swimming speed, which might further impact their mating, predation and feeding.

We note that the present study considered only spherical swimmers. Non-spherical plankton, such as elongated ones, probably experience a fluid inertial torque stemming from both their non-spherical shape (Dabade et al., 2015; Sheikh et al., 2020; Gustavsson et al., 2019; Qiu et al., 2022a) and propulsion mechanism (Candelier et al., 2022). While the analytical solution

for the fluid inertial torque on a non-spherical swimmer remains unclear, fully resolved numerical simulation could be used to reveal the dynamics of non-spherical settling swimmers. The resulting findings could be potentially applied to the model of point-like swimmer.

*Code and data availability.*  Raw data from simulation are available upon request to corresponding author.

*Author contributions.*  J.Q., E.C and L.Z. designed the project; J.Q. and L.Z. performed research; J.Q. and Z.C. developed numerical tools;

J.Q. analyzed data; J.Q., E.C. and L.Z wrote the paper.

*Competing interests.*  The authors declare no conflict of interest.

*Acknowledgements.*  This work was supported by the National Natural Science Foundation of China (Grant No. 92252104 and 92252204). Z.C. acknowledges the support by China Postdoctoral Science Foundation (grant number 2022M721849).



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
