# Peer review of "Clustering of settling microswimmers in turbulence"

_EGUsphere, 2023_

## Author Comment (AC1)

**Referee's General Opinion:**

This paper discusses the statistical properties of settling microswimmers in turbulence. In the first part the mathematical model for settling swimmers is derived and in the second part the results of numerical simulations in a turbulent flow are discussed.

I am not convinced that the present version of the manuscript deserves publication.

**Author's Reply:**

We appreciate the careful reading of our manuscript and your constructive suggestions for improving the quality of the paper. In response to your major comments, we provide clarifications below and additional text is clearly marked in document. We hope that the revised version will be suitable for publication.

**Referee's Comment #1**

The derivation of the model (section 2.1) is interesting and its study probably deserves publication. The main objection I have is that in the limit discussed in this paper (eqs. 5-8). the model is identical to the Kessler model for gyrotaxis which has been already studied in the same flow and with the same statistical approach (clustering in terms of Voronoi distribution and preferential concentration). Therefore, it is not clear what are the new results of this manuscript.

Moreover, this is not clearly discussed in the paper. After eqs. (5-8), it is written that "the second term on the rhs of (8) is analogous to the gyrotactic effect", while the model (6,8) is identical to the standard gyrotactic model.

**Author's Reply:**

One of your criticisms is that, in the limit of our paper, our model is identical to the Kessler model for gyrotaxis. However, despite the similarities in mathematical forms, the mechanism responsible for the orientation effect is totally different from that in Kessler's model. In the Kessler model, the reorientation effect is caused by the offset between the mass center and the shape center of a plankton cell. The magnitude of the gravitational reorientation torque is determined by the distance of the offset as well as some other physical properties such as fluid viscosity and density ratio, etc. As a result, the Kessler reorientation torque is a physical mechanism regardless of whether a plankton is swimming or settling or not. In our model, however, the reorientation effect is caused by the fluid inertial torque when a plankton swim and settling relative to the ambient fluid, and thus depends on the swimming and settling speeds. The magnitude of the reorientation angular velocity is no longer an independent parameter, which is drastically different from the case of Kessler model.

We believe this difference is important and deserves investigation. First, it allows a plankton to modify its swimming speed and then control its reorientation behavior. Second, our results suggest that the settling speed also matters even if a plankton is motile, allowing fluid inertial torque to generate a reorientation effect. To address both points, we studied how clustering and preferential sampling are altered when micro-swimmers have different swimming and settling speeds ($\Phi_s$ and $\Phi_g$). To emphasize the dependency of the reorientation effect on the swimming and settling speeds, we discuss the clustering and preferential sampling in a $\Phi_s$ vs $\Phi_g$ parameter diagram in the paper. For instance, we show that clustering intensity is the largest when $\Phi_s = 10$ and $\Phi_g =$

0.5. This was not observed in earlier studies where fluid inertial torque and the settling speed are not considered as a mechanism of reorientation effect.

To address the difference between our model and the Kessler model, we have added a discussion in section 2.1:

Around line 90:

*"The last term of Eq.(8) indicates that fluid inertial torque drives a squirmer swimmer to swim against gravity. Here, we use a dimensionless timescale $\Psi_I$ to quantify the effect of fluid inertial torque. $\Psi_I$ can be understood as the dimensionless time that a swimmer in still fluid restores upward orientation from an inclined orientation under a reorientation torque. This is analogous to the gyrotactic effect induced by bottom-heaviness, which is typically expressed as $2\Psi^{-1}(e_g \times n)$ (Kessler, 1986). We note that, however, they are two different mechanisms. The torque generated by bottom-heaviness depends on the distance of the offset between the center of gravity and hydrodynamic forces on a cell, which is usually determined by morphology. On the contrary, fluid inertial torque depends on the swimming and settling speeds and, determined by motility"*

**Referee's Comment #2**

In conclusion, I think that it would be interesting to investigate the model general model (3,4) and compare it with the known gyrotactic limit. This would add something new to our understanding to swimming microorganisms in turbulence. Moreover, my impression is that, with the typical values discussed after (9), the range of the Stokes numbers is comparable with the other dimensionless parameters and therefore the limit St->0 is not justified.

**Author's Reply:**

We appreciate your suggestion that we can study the original model (3) and (4) to compare with the Kessler model. It would be interesting to see whether new physical phenomena emerge when the Stokes number *St* is not negligibly small. However, we want to focus the scope of the current paper on plankton or other small microorganisms in water. Based on their typical physical properties, the Stokes number approaches to zero. For instance, we assume a plankton cell has typical size $a = 0.1\,\text{mm}$ and it is 5% heavier than the water. For the flow environment, we use the parameter range in the main text after Eq. (9). The range of dissipation rate cited in our text gives the range of Kolmogorov time scale for ocean turbulence, $\tau_f$= 31.6 to 1.0 s. At last, using the definition of $St = 2 * 1.05a^2/(9\gamma\tau_f)$ , St number ranges from 0.0001 to 0.0023. This is much smaller than other dimensionless parameters such as $\Phi_s$ and $\Phi_g$ in our model. A similar but more detailed estimate of dimensionless parameters can be referred to Table II in our recent paper (Qiu et al., Physical Review Research 4, 023094 (2022)), which is attached below. As a result, it is more appropriate to focus on the limit of zero Stokes number for the present study.

To address this issue, we have added discussions in section 2.1 in the main text to demonstrate the range of Stokes number to validate our simplification.

Around Line 80:

"Typically, St of planktonic microswimmers are usually negligibly small as summarized in Qiu et al. (2022a). For instance, using $a = 0.1mm$, $\rho_p/\rho_f = 1.05$, and using typical turbulence Kolmogorov timescale $\tau_f = 31.6\ to\ 1.0\ s$, one obtains $St = 1.0 \times 10^{-4}\ to\ 2.3 \times 10^{-3}$."

TABLE II. Dimensionless numbers of typical plankton species shown in Table I. The Kolmogorov scales of ocean turbulence is calculated with $\gamma = 1.058 \times 10^{-6}$ m$^2$ s$^{-1}$, and the energy dissipation rate $\epsilon$ ranges from $1 \times 10^{-9}$ to $1 \times 10^{-6}$ m$^2$ s$^{-3}$ [42]. Re$_p$ is calculated with Re$_p = v_{\text{swim}}L/\gamma$ because $v_{\text{swim}} > V_{\text{settle}}$ for many species in Table I. Superscript[1]: The values are calculated with $\lambda = 2.3$ similar to *Centropages typicus*. Superscript[2]: The values are calculated with $\lambda = 2.0$.

| Species | | $\Phi_{\text{swim}}$ | $\Phi_{\text{settle}}$ | $M$ | Re$_p$ | St $(\times 10^{-5})$ | $\Psi_I$ |
|---|---|---|---|---|---|---|---|
| *Cochlodinium polykrikoides* [39] | Single cell | 2.17~0.39 | 0.14~0.03 | −0.078 | 0.015 | 0.17~5.52 | 20.7~655.2 |
| | 2-cells | 3.32~0.59 | 0.1~0.03 | −0.101 | 0.029 | 0.22~6.95 | 9.1~287.2 |
| | 4-cells | 4.44~0.79 | 0.23~0.04 | −0.136 | 0.077 | 0.45~14.11 | 3.6~112.5 |
| | 8-cells | 4.75~0.84 | 0.36~0.06 | −0.137 | 0.147 | 0.90~28.51 | 2.1~67.4 |
| *Centropages typicus* [28,41] | early nauplius | 1.83~0.33 | 0.28~0.05 | −0.114 | 0.041 | 1.29~40.78 | 8.7~274.2 |
| | late nauplius | 3.99~0.71 | 0.78~0.14 | −0.114 | 0.153 | 3.75~118.48 | 1.4~44.9 |
| *Euterpina acutifrons* [28] | late nauplius[1] | 5.99~1.06 | 1.44~0.26 | −0.114 | 0.204 | 2.96~93.61 | 0.5~16.1 |
| *Eurytemora affinis* [28] | late nauplius[1] | 9.09~1.62 | 1.01~0.18 | −0.114 | 0.313 | 3.02~95.49 | 0.5~15.2 |
| *Temora longicornis* [28,40] | late nauplius[1] | 3.16~0.56 | 1.33~0.24 | −0.114 | 0.166 | 7.02~222.01 | 1.0~33.1 |
| | copepod[1] | 4.55~0.81 | 0.94~0.17 | −0.114 | 0.231 | 6.57~207.83 | 1.0~32.5 |
| *Ceratium tripos*[2] [27] | | 0.93~0.16 | 0.91~0.16 | −0.101 | 0.012 | 0.46~14.60 | 5.9~186.1 |
| *Ceratium furca*[2] [27] | | 4.32~0.77 | 0.34~0.06 | −0.101 | 0.033 | 0.17~5.50 | 3.3~105.9 |
| *Akashiwo sanguinea*[2] [27] | | 1.66~0.30 | 0.30~0.05 | −0.101 | 0.012 | 0.15~4.81 | 9.9~314.3 |
| *Dinophysis acuminata*[2] [27] | | 1.84~0.33 | 0.18~0.03 | −0.101 | 0.010 | 0.09~2.84 | 15.2~481.8 |
| *Alexandrium minutum*[2] [27] | | 1.54~0.27 | 0.05~0.01 | −0.101 | 0.005 | 0.03~0.89 | 58.3~1843.6 |
| *Prorocentrum minimum*[2] [27] | | 1.14~0.20 | 0.03~0.00 | −0.101 | 0.002 | 0.01~0.44 | 159.8~5053.4 |

Table II in Qiu et al., Physical Review Research 4, 023094 (2022). In this table, $\Phi_{\text{swim}}$ and $\Phi_{\text{settle}}$ means $\Phi_s$ and $\Phi_g$ in the present paper. We also note that $\Psi_I$ in this table does not refers to the reorientation time scale defined in the present paper.

Referee's Comment #3

Minor point: the presentation of the model and the results is not always clear. For example, the settling speed v_g is not defined.

Author's Reply:

Thank you. We remove the symbol v_g and give the definition of $\Phi_g$ directly, $\Phi_g = 2(\rho_p/\rho_f - 1)a^2 g/(9\gamma u_f)$. It can be found right below Eq. (4).

---

## Author Comment (AC2)

As the authors point out, there have been a number of papers examining clustering of microswimmers due to gyrotaxis in homogeneous isotropic turbulence over the past one or two decades.

The main contribution of the present paper appears to be the addition of the inertial torque (second term on the RHS of Eq. 2) to an otherwise standard model of a microswimmer that has previously been studied. This inertial torque comes from the work of Candelier et al. (2022). Candelier et al. go to great lengths to describe the various orders of approximations in their work and the conditions under which their results are valid. However, the authors of the present paper do not clarify whether the situation they consider is consistent with including this extra term and whether they can include only this extra term without the need to include other extra terms for consistency. In a similar vein, the authors (between Eq 4 and Eq 5) state the microswimmer inertia is negligible and consider the limit where their St —> 0 without any explanation of whether the limit exits at finite Re (required for the inertial torque to be relevant).

**Author's Reply:**

Thank you for your constructive comments. One of your major comments is concerned with the validation of our model in the situation currently discussed. We apologize for the lack of clarification of model assumptions. We provide a clearer clarification as follows about how the assumptions of our model are met in the parameter range that is specified right after Eq. (9).

The first assumption is that we adopted Candelier's model for the fluid inertial torque. This requires the Reynolds number to be finite but much smaller than unity, which is satisfied for typical marine microorganisms. For instance, we assume the radius and the swimming speed of a typical plankton to be a = 0.1 mm and $v_s$ = 0.5 mm/s, respectively. The obtained Reynolds number is $Re = 2av_s/\gamma = 0.1$, where $\gamma = 1\,mm^2/s$ is the kinematic viscosity of fluid. The Candelier's model is expected to describe the inertial torque quite precisely at such a low Reynolds number according to Candelier et al. (2022). A more detailed presentation of Reynolds number of typical plankton species can be referred to Table II in our earlier publication (Qiu et al., Physical Review Research 4, 023094 (2022)), which is shown below.

The second assumption is made when we derive Eqs. (5-8) from Eqs. (3) and (4). Here, we assume the Stokes number is negligibly small. The Stokes number, usually used to quantifying the inertia of a rigid particle, is defined as $St = 2Da^2/(9\gamma\tau_f)$, where a, D, and $\tau_f$ are the radius of the plankton, the cell-to-fluid density ratio, and the time scale of fluid motion, respectively. For typical plankton parameters, we have a = 0.1 mm and D = 1.05. Considering the typical intensity of ocean turbulence, $\tau_f$ ranges from 1.0 to 31.6 s (Kiørboe & Enric 1995). Using all these parameters, we estimate that St ranges from 0.0001 to 0.0023. With such small St, our derivation is justified.

At the limit of zero St, the fluid inertial torque is not negligible as long as the swimming and settling speeds are not so small. This is shown in Eq. (4). When St ->0, the last term on the right-hand-side, which represents fluid inertial torque, is unaffected. Its magnitude is determined by a coefficient scaling with $\tau_f u_f^2/\gamma$ as well as the dimensionless swimming and settling speeds, $\Phi_s$ and $\Phi_g$. In a turbulent flow, $\tau_f$ and $u_f$

are chosen as the Kolmogorov time and velocity scales, giving $\tau_f\, u_f^2\, /\, \gamma = 1$. This equation is given by the definition of Kolmogorov scales, where $\tau_f = (\gamma/\epsilon)^{1/2}$, $u_f = (\gamma\epsilon)^{1/4}$, and $\epsilon$ is the energy dissipation rate of turbulence. As a result, the fluid inertial term is comparable to other terms as long as the dimensionless swimming and settling speeds ($\Phi_s$ and $\Phi_g$) are finite. According to the Table below, both $\Phi_s$ and $\Phi_g$ vary across a wide range depending on species and flow condition. For some zooplankton species, $\Phi_s$ and $\Phi_g$ are large enough so that the inertial torque is not negligible, while Re and St are still within the range of our model assumptions.

To address the validation of our model, we revised the text in section 2.1 (additional text is clearly marked), and we show typical values of St and Re using physical properties of marine plankton.

Around line 55:

*"The motion of plankton in fluid flows is usually described by a micro-swimmer model (citations, ⋯), which assumes a plankton to be a point-like micro-swimmer carried by a fluid flow. This assumption is justified when the Reynolds number, $Re = a|\boldsymbol{v} - \boldsymbol{u}|/\gamma$, is much smaller than unity. Here, the Reynolds number is defined based on the radius of a swimmer, $a$, the differences between the velocities of a swimmer $\boldsymbol{v}$ and its ambient undisturbed flow $\boldsymbol{u}$, and the kinematic viscosity of the fluid $\gamma$. For typical plankton species, this assumption is justified because of their tiny size and limited motility, as summarized in our recent publication (Qiu et al., 2022a). For instance, the typical size and swimming speed of zooplankton are $a = 0.1\, mm$, $|\boldsymbol{v} - \boldsymbol{u}| = 1.0\, mm$, respectively. Accordingly, we obtain $Re = 0.1$ using the viscosity of water $\gamma = 10^{-6} m^2/s$"*

Around line 70:

*"The model of fluid inertial torque is derived in the limit of Re → 0, but it has been shown to be justified when Re < 0.3 (Candelier et al., 2022), within the typical range of plankton physical properties (Qiu et al., 2022a)"*

Around Line 80:

*"Typically, St of planktonic microswimmers are usually negligibly small as summarized in Qiu et al. (2022a). For instance, using $a = 0.1 mm$, $\rho_p/\rho_f = 1.05$, and using typical turbulence Kolmogorov timescale $\tau_f = 31.6\, to\, 1.0\, s$, one obtains $St = 1.0 \times 10^{-4}\, to\, 2.3 \times 10^{-3}$."*

TABLE II. Dimensionless numbers of typical plankton species shown in Table I. The Kolmogorov scales of ocean turbulence is calculated with $\gamma = 1.058 \times 10^{-6}$ m$^2$ s$^{-1}$, and the energy dissipation rate $\epsilon$ ranges from $1 \times 10^{-9}$ to $1 \times 10^{-6}$ m$^2$ s$^{-3}$ [42]. Re$_p$ is calculated with Re$_p = v_{swim}L/\gamma$ because $v_{swim} > V_{settle}$ for many species in Table I. Superscript[1]: The values are calculated with $\lambda = 2.3$ similar to *Centropages typicus*. Superscript[2]: The values are calculated with $\lambda = 2.0$.

| Species | | $\Phi_{swim}$ | $\Phi_{settle}$ | $M$ | Re$_p$ | St $(\times 10^{-5})$ | $\Psi_I$ |
|---|---|---|---|---|---|---|---|
| *Cochlodinium polykrikoides* [39] | Single cell | 2.17~0.39 | 0.14~0.03 | −0.078 | 0.015 | 0.17~5.52 | 20.7~655.2 |
| | 2-cells | 3.32~0.59 | 0.1~0.03 | −0.101 | 0.029 | 0.22~6.95 | 9.1~287.2 |
| | 4-cells | 4.44~0.79 | 0.23~0.04 | −0.136 | 0.077 | 0.45~14.11 | 3.6~112.5 |
| | 8-cells | 4.75~0.84 | 0.36~0.06 | −0.137 | 0.147 | 0.90~28.51 | 2.1~67.4 |
| *Centropages typicus* [28,41] | early nauplius | 1.83~0.33 | 0.28~0.05 | −0.114 | 0.041 | 1.29~40.78 | 8.7~274.2 |
| | late nauplius | 3.99~0.71 | 0.78~0.14 | −0.114 | 0.153 | 3.75~118.48 | 1.4~44.9 |
| *Euterpina acutifrons* [28] | late nauplius[1] | 5.99~1.06 | 1.44~0.26 | −0.114 | 0.204 | 2.96~93.61 | 0.5~16.1 |
| *Eurytemora affinis* [28] | late nauplius[1] | 9.09~1.62 | 1.01~0.18 | −0.114 | 0.313 | 3.02~95.49 | 0.5~15.2 |
| *Temora longicornis* [28,40] | late nauplius[1] | 3.16~0.56 | 1.33~0.24 | −0.114 | 0.166 | 7.02~222.01 | 1.0~33.1 |
| | copepod[1] | 4.55~0.81 | 0.94~0.17 | −0.114 | 0.231 | 6.57~207.83 | 1.0~32.5 |
| *Ceratium tripos*[2] [27] | | 0.93~0.16 | 0.91~0.16 | −0.101 | 0.012 | 0.46~14.60 | 5.9~186.1 |
| *Ceratium furca*[2] [27] | | 4.32~0.77 | 0.34~0.06 | −0.101 | 0.033 | 0.17~5.50 | 3.3~105.9 |
| *Akashiwo sanguinea*[2] [27] | | 1.66~0.30 | 0.30~0.05 | −0.101 | 0.012 | 0.15~4.81 | 9.9~314.3 |
| *Dinophysis acuminata*[2] [27] | | 1.84~0.33 | 0.18~0.03 | −0.101 | 0.010 | 0.09~2.84 | 15.2~481.8 |
| *Alexandrium minutum*[2] [27] | | 1.54~0.27 | 0.05~0.01 | −0.101 | 0.005 | 0.03~0.89 | 58.3~1843.6 |
| *Prorocentrum minimum*[2] [27] | | 1.14~0.20 | 0.03~0.00 | −0.101 | 0.002 | 0.01~0.44 | 159.8~5053.4 |

Table II in Qiu et al., Physical Review Research 4, 023094 (2022). In this table, $\Phi_{swim}$ and $\Phi_{settle}$ means $\Phi_s$ and $\Phi_g$ in the present paper. We also note that $\Psi_I$ in this table does not refers to the reorientation time scale defined in the present paper.

**Referee's Comment #2**

The final microswimmer equations used in the simulations (Eq 5 — Eq 8) are dubious for the reasons outlined above. However, taken them as a given, the results are not particularly novel because these microswimmer equations simplify to those investigated before (spherical gyrotactic settling microswimmers) and thus the results are not particularly novel.

**Author's Reply:**

You pointed out that our model simplifies to the classic Kessler's gyrotaxis model. However, despite the similarities in mathematical forms, the mechanism responsible for the orientation effect is totally different from that in Kessler's model. The fluid inertial torque makes a swimmer to swim in upward direction because the symmetry of flow ambient field is broken by the swimming behavior. In Kessler's model, a swimmer swim upwards under a gravity torque due to the offset between the centers of mass and the hydrodynamic force. The difference in the mechanisms can be seen in the definition of the reorientation time scale $\Psi$. In our model, $\Psi_I$ depends on both swimming and settling speeds, while in Kessler's model, $\Psi$ depends on the size of swimmer and the distance of the offset mentioned above.

We believe this difference is important and deserves investigation. First, it allows a plankton to modify its swimming speed and then control its reorientation behavior. Second, our results suggest that the settling speed also matters when we consider a motile plankton, because it generates a reorientation effect under fluid inertial. To address both points, we studied how clustering and preferential sampling are altered when micro-swimmers have different swimming and settling speeds.

This paper aims to understand the influence of fluid inertial torque in the reorientation phenomenon of micro-swimmers. Therefore, we isolate the term for fluid inertial torque in Eq. (4), which simplifies the discussion. To model the motion of an actual

plankton precisely, one must also consider other relevant mechanisms such as bottom-heaviness or phototaxis. However, they are not in the scope of this paper.

We have revised section 2.1 to address the difference between our model and Kessler's model.

Around line 90:

*"This is analogous to the gyrotactic effect induced by bottom-heaviness, which is typically expressed as $2\Psi^{-1}(e_g \times n)$ (Kessler, 1986). We note that, however, they are two different mechanisms. The torque generated by bottom-heaviness depends on the distance of the offset between the center of gravity and hydrodynamic forces on a cell, which is usually determined by morphology. On the contrary, fluid inertial torque depends on the swimming and settling speeds and, determined by motility"*

**Referee's Comment #3**
Additionally, the authors apply their model to extremely high swimming speeds and high settling speeds (L91) without any comment on whether this range of values are consistent with their assumptions. (I suspect they are not).

**Author's Reply:**
As shown in the Table above, large $\Phi_s$ and $\Phi_g$ can be reached by zooplankton species in weak turbulence, while Re and St are still within the range of our model assumptions. For instance, the nauplius of *Eurytemora affinis* has $\Phi_s$ = 9.1, $\Phi_g$ = 1.0, Re = 0.3 and St = $3*10^{-5}$.

To address our assumption about the range of dimensionless parameters, we added some sentences to discuss the typical parameters of marine planktons.

Around Line 105:

*"Large $\Phi_s$ and $\Phi_g$ are reached by swimmers with strong motility in weak turbulence which $u_\eta$ is small. In such case, the assumptions of our model are still justified. First, Re can be still small even for plankton that swim fast as long as their size is sufficiently small. Second, St is independent of plankton's motility, which has been shown to be negligibly small for typical turbulence conditions in the ocean (Qiu et al., 2022a)"*

**Referee's Comment #3**
Minor comments:
L16, 'accumulates' — check grammar
L17, 'clustering' — check grammar
L22, 'the inverse of a timescale B' — unclear writing. It needs to be explained here what B is (timescale for reorientation against gravity in an otherwise quiescent environment).
Eq. 4 — v_s\prime should be \Phi_s

**Author's Reply:**
We thank you for your careful review. We have addressed these points in the revised manuscript.

**Referee's Comment #4**
L146, 'variance of Voronoi volumes' — I'm not sure whether the authors mean the location of the peak of the distribution or indeed the variance.
**Author's Reply:**
The term 'variance of Voronoi volumes' means the actual variance of the Voronoi volumes of each swimmer. To avoid ambiguity, we explain the term when it appears in the text first time.

Around line 160:

"We calculate the Voronoi volume of each swimmer, and obtain the variance of volume distribution normalized by the mean volume of each swimmer, $\sigma_{\hat{V}}^2 = E\left(\frac{V}{<V>} - 1\right)^2$ .

---

## Author Response (AR2)

**Overall comment:**

This article aims at studying a model system for motile spherical plankton that takes into account the torque force induced by the fluid inertia (a force that was only recently derived by Candelier et al. 2022). When the assumption of small Stokes number is applied in such a setting, the model greatly simplifies and becomes a special case of the Kessler (1986) model of gyrotaxis. This implies that its dynamic is coincident with the one already known. In fact the gyrotactic model has been studied in the past with the same techniques here adopted, i.e. Voronoi distribution and preferential concentration. It is such special limit that is investigated here by the authors.

The coincidence between the present and the existing classical gyrotactic model has been pointed out by both the previous reviewers, which also requested for a more accurate justification of the limit (small Stokes numbers) addressed in this study.

In their rebuttal letter the authors acknowledge the de facto identity between the present model and Kessler's one. However, they make the point that the physical mechanisms at their origin are different, and that they might have different biological implications. Furthermore, the assumption of the small Stokes is discussed and supported in a quantitative way.

In my view these exchanges with the reviewers and the resulting revised manuscript, highlight a scientific discussion that deserves to be published, as it goes beyond the mere topic of the originality of the model.

I am therefore favourable to the publication of this manuscript, with the following minor modifications.

**Authors' reply:**

We appreciate the referee's valuable evaluation of our work and we are pleased to learn that the referee considers our paper publishable. Our reply to the referee's comments is listed below. Any new and/or reformulated pieces of text are highlighted in red in the revised manuscript.

**Comment #1:**

1) In order to remove any ambiguity, still present in the revised manuscript, concerning the link between this model and the one of Kessler, I suggest to amend the following sentences:

Around line 90 "This is analogous to the gyrotactic effect induced by bottom-heaviness, which is typically expressed as ... (Kessler,1986)."
"analogous" -> identical to

Line 137 "As discussed earlier, the fluid inertial torque on a settling swimmer induces an effect similar to gyrotaxis mechanism."
"similar to" -> equivalent to, coincident with

Line 158 : "This is analogous to the effect of bottom-heaviness (Kessler,1986), which also drives swimmers to orientate upward with a time scale $\Psi$ dependent on the offset of the center of gravity with the center of hydrodynamic forces."
"analogous to" - > the same as

line 180 : "This observation is similar to Durham et al.(2013) where the maximal preferential

sampling is also reached when $\Psi \approx 1$.

"is similar to " -> clearly agrees with

**Authors' reply:**

We thank the referee for pointing out the issues that should be amended. We have addressed the referee's comments to ensure the connection between our and Kessler's models is correctly stated.

**Comment #2:**

2) It is obvious that the complete model Eq. (1)-(2) would deserve a future study. I suggest that this is mentioned in the conclusions of the manuscript.

**Authors' reply:**

We have addressed the referee's comment by adding a paragraph in the section of conclusions to discuss the necessary of considering the full dynamics of swimmers in future studies.